# The Neuropathological Diagnosis of Alzheimer’s Disease—The Challenges of Pathological Mimics and Concomitant Pathology

**DOI:** 10.3390/brainsci10080479

**Published:** 2020-07-24

**Authors:** Andrew King, Istvan Bodi, Claire Troakes

**Affiliations:** 1Department of Clinical Neuropathology, King’s College Hospital, Denmark Hill, London SE5 9RS, UK; istvan.bodi@nhs.net; 2London Neurodegenerative Diseases Brain Bank, Institute of Psychiatry, Psychology & Neuroscience, King’s College, London SE5 8AF, UK; claire.troakes@kcl.ac.uk; 3Department of Basic and Clinical Neuroscience, Institute of Psychiatry, Psychology Neuroscience, Kings College London, London SE5 9RT, UK

**Keywords:** Alzheimer’s, dementia, TDP-43, Lewy, vascular, tauopathies

## Abstract

The definitive diagnosis of Alzheimer’s disease (AD) rests with post-mortem neuropathology despite the advent of more sensitive scanning and the search for reliable biomarkers. Even though the classic neuropathological features of AD have been known for many years, it was only relatively recently that more sensitive immunohistochemistry for amyloid beta (Aβ) and hyperphosphorylated tau (HP-tau) replaced silver-staining techniques. However, immunohistochemistry against these and other proteins has not only allowed a more scientific evaluation of the pathology of AD but also revealed some mimics of HP-tau pathological patterns of AD, including age-related changes, argyrophilic grain disease and chronic traumatic encephalopathy. It also highlighted a number of cases of AD with significant additional pathology including Lewy bodies, phosphorylated TDP-43 (p-TDP-43) positive neuronal cytoplasmic inclusions and vascular pathology. This concomitant pathology can cause a number of challenges including the evaluation of the significance of each pathological entity in the make-up of the clinical symptoms, and the threshold of each individual pathology to cause dementia. It also raises the possibility of underlying common aetiologies. Furthermore, the concomitant pathologies could provide explanations as to the relative failure of clinical trials of anti-Aβ therapy in AD patients.

## 1. Introduction

Alzheimer’s disease is the most common cause of dementia in the Western world and is especially prevalent after 60 years of age. The final definitive diagnosis rests with the neuropathology and the classic neuropathological features of neuritic plaques and neurofibrillary tangles, together with neuronal loss have been recognised for over a century. Although the main pathological proteins seen within the brains of AD patients are amyloid beta (Aβ) and hyperphosphorylated tau (HP-tau) it is still not fully established as to the relationship between these two proteins. The amyloid cascade hypothesis first proposed by Hardy et al., argued that Aβ was the initial driving force of the disease and that the accumulation of HP-tau in the form of neurofibrillary tangles, neurites and neuropil threads was a secondary event [1]. This was supported not least by the fact that patients with mutations in beta amyloid precursor protein (β-APP) (which is a precursor for Aβ), including patients with Down’s syndrome (β-APP is located on chromosome 21), developed AD, whereas patients with tau mutations developed a spectrum of different neurodegenerative disorders, often with parkinsonian symptoms. The problem with this hypothesis from a neuropathological perspective is that the degree and extent of tau pathology in AD patients’ brains correlated much better with the patients’ cognitive symptoms in life than the degree of Aβ pathology [2,3]. It is now considered that the Aβ seen on pathological slides may represent the tip of the iceberg and it is the soluble oligomeric form that causes neuronal damage [4]. Despite the widespread acceptance of the amyloid cascade hypothesis, anti-Aβ therapy such as solanezumab and verubecestat, although showing encouraging effects on reducing Aβ load in animals, have had only disappointing results in human trials [5]. The relative failure of this line of treatment, together with the associated enormous cost of drug development, has led to some pharmaceutical companies withdrawing from neurodegenerative research altogether. Although the cohorts in the trials were in the group of mild-to-moderate AD, it could be argued that the subjects were still too demented for treatment to be likely to be effective. Another reason could be that the target engagement of the therapy was never entirely established [6]. Others have called into question the amyloid cascade hypothesis, and instead proposed a tau first hypothesis [7]. There is, however, another possible explanation for this treatment failure. Recently, with the greater range of available antibodies and greater sampling of post-mortem brains, it has been shown that many patients with typical clinical AD features actually have not only the characteristic Aβ and HP-tau deposition in the form of plaques and tangles, respectively, but also additional pathology. This includes infarcts or other cerebrovascular disease pathology, Lewy bodies, and TDP-43 pathology. It is not unreasonable to assume that this additional pathology may be one of the reasons that the trials have been relatively unsuccessful. The presence of an additional pathology may also complicate the development of proposed biomarkers of AD. The purpose of this review is to highlight some of the challenges to AD neuropathological diagnosis including some of the pathological “mimics” and some of the other additional pathologies which may affect the clinical diagnosis. It also aims to provide some practical issues to assist with overall neuropathological diagnosis.

## 2. The Neuropathological Diagnosis of Alzheimer’s Disease 

Considering that AD is relatively common and the main neuropathological features have been recognised for over a century, one would have thought that the diagnosis of AD would be relatively straightforward. However, the introduction of immunohistochemical techniques and the greater sampling of brain regions, together with the recognition of specific “age-related” changes has, if anything, made the neuropathological diagnosis much more challenging in recent years. Often, the post-mortem brains reveal marked cerebral atrophy (Figure 1a), and originally the neuropathological diagnosis of AD depended on the presence of “senile” or “neuritic” plaques and “neurofibrillary tangles“ detected by silver stains. This technique was commonly used until the 1990s, when reliable immunohistochemistry was established. This immunohistochemistry revolutionised the diagnostic potential and also allowed a somewhat more scientific study of the pathological proteins involved in the disease process. The main antibodies employed are against Amyloid Beta (Aβ), which highlights plaques (Figure 1b,f) (together with amyloid in the walls of vessels-cerebral amyloid angiopathy (CAA)), and hyperphosphorylated (HP)-tau in the form of neuritic plaques (Figure 1c), neurofibrillary tangles (Figure 1d), and neuropil threads (Figure 1c–e). Because the disease was thought to pass through a number of stages, it was never going to be sufficient to make a diagnosis of Alzheimer’s disease on 1 or 2 brain blocks; instead, a staging protocol had to be established. It was thought that the stages went from entorhinal to limbic and then isocortical. One of the earliest accepted staging systems for AD concentrated on neurofibrillary tangle pathology and neuropil threads, and was first devised by Braak and Braak using silver staining; stages 0-VI were designated, and they found good clinical correlation [8] (Appendix A). It was later perfected and “tweaked” to be used with the development of tau immunohistochemistry, and the Brain Net Europe (BNE) Consortium established a wide sampling technique to apply this further, especially with regard to the density of neuropil threads (Appendix A) [9]. The neuritic plaque element to staging has always been more problematic. It has remained relatively unchanged since the 1990s and rested on the semi-quantitative score of neuritic plaques in a defined number of cerebral areas [10]. Although originally it relied on silver staining techniques in which neuritic plaques can be readily identified, it was later suggested that Aβ be employed, and although the so-called “cored” Aβ plaques are thought to most likely represent neuritic plaques, the antibody cannot detect the actual “neuritic” component, and therefore HP-tau is probably a better marker. This assessment then gives an age-related score of the density of the plaques (Appendix A). However, although the designated blocks are still evaluated, the age-related element is now considered to be less important and generally an assessment of absent, mild, moderate and severe (C0-C3) plaque density is made. The last component in the staging system is the extent of Aβ parenchymal pathology throughout the brain, and this was established by Thal et al. [11]. The stages are called phases and are numbered 0–5 (Appendix A). In order to develop a system that combined all these above pathological features, the National Institute on Aging/Alzheimer’s Association (NIA-AA) established scores comprising the extent of Aβ pathology, (A), Braak or modified Braak score, (B) and CERAD plaque score (C) (Table 1) and, from this combination, gave a likelihood of the degree of AD neuropathological change in an individual case (Table 2) [12,13]. The whole process of the neuropathological diagnosis can be followed along the pathways of Figure 2. Therefore, having finally established the degree of AD neuropathological changes in a case, there are a number of other factors to consider before coming to an overall final diagnosis. These include some pathology mimicking features of AD, e.g., different types of tauopathy such as primary age-related tauopathy (PART), age-related tau-astrogliopathy, (ARTAG), and argyrophilic grain disease (AGD). All of these pathological diagnoses can cause confusion and sometimes they can co-exist with AD, especially ARTAG, adding to the diagnostic difficulties. The final major difficulty is trying to determine the presence, extent and significance of any concomitant pathological processes. 

### 2.1. Neuropathological Entities Mimicking Aspects of Alzheimer’s Disease 

#### 2.1.1. Primary Age-Related Tauopathy (PART)

Primary age-related tauopathy (PART) is a relatively newly defined entity constructed to define a pathological continuum ranging from focally distributed neurofibrillary tangles seen in cognitively normal aged individuals to what has been described previously as tangle only dementia, and senile dementia of the neurofibrillary tangle type [14]. The criteria include AD-type neuropathological changes (Figure 3a) without, or with few, Aβ plaques (maximum Thal phase 2) and a lack of neuritic plaques. There are categories of definite PART when there is an associated Thal Aβ phase of 0, and possible PART when there is an associated Thal Aβ phase of 1–2. By definition, the Braak Neurofibrillary tangle stage must be less than or equal to IV, and is usually III or lower. This, of course, leads to difficulties in diagnosing neuropathologically since there is necessarily overlap with AD, but the entity was primarily designed to differentiate those cases with very little Aβ pathology, especially for communication between neuropathologists and across research circles. Whether it truly represents a distinct disease completely separate from AD awaits further clinical, genetic and biochemical evaluation.

##### Practical Considerations 

From a practical perspective, the main issue in this diagnosis is the presence of neurofibrillary tangles, and neuropil threads, which may be quite dense in areas but are associated with little or no Aβ pathology and no neuritic plaques. Therefore, it is essential that there has been extensive sampling of the brain and staining of many blocks for Aβ and HP-tau in order to rule out a possible diagnosis of AD. 

#### 2.1.2. Aging-Related Tau Astrogliopathy (ARTAG) 

Aging-related tau astrogliopathy (ARTAG) is recently introduced terminology to describe age-related tauopathy changes within astroglial cells that are distinct from the glial tau pathology seen in well-characterised neurodegenerative diseases such as Pick’s disease (PiD), corticobasal degeneration (CBD), progressive supranuclear palsy (PSP), argyrophilic grain disease (AGD) and globular glial tauopathy (GGT) [15,16,17,18]. These include “thorny astrocytes” (Figure 3b,c) and (c) inset, and fine granular tau immunoreactivity in astrocytic processes, the so-called “granular/fuzzy astrocytes” (Figure 3d). As opposed to the diseases above, the astrocytic tau pathology in ARTAG is seen in the superficial cortex including subpial regions (Figure 3b), and subependymal, and perivascular (Figure 3c) distributions, as well as in white matter, most frequently in the limbic and medial temporal areas. It can be seen without or in addition to AD pathological features, and can sometimes complicate the staging of AD. There are two main groups of tau: 3-repeat (3-R) tau, and 4-repeat (4-R) tau (formed by alternative slicing of the microtubular-associated protein tau (MAPT) gene). Some diseases such as PiD are 3-R tauopathies, whilst others such as PSP, CBD, AGD and GGT are 4-R tauopathies, whereas AD has both 3-R and 4-R tau deposits. ARTAG is considered to be a 4-R tauopathy since 4-R tau (and not 3-R tau) is seen in the glial cells and can be stained for, but biochemically there is a mixture of 3-R and 4-R tau, which may be a result of contamination by co-existing AD pathological structures. 

##### Practical Considerations

It is obviously important in such cases to have an indication of the age of the subject and the degree of cognitive decline. When unaccompanied by other pathology, the pathological features of ARTAG can be straightforward, with the characteristic perivascular, subpial and perivascular astrocytic tau pathology. However, it can become much more difficult in the background of another tauopathy including AD. In such cases, it may sometimes be necessary to employ immunohistochemistry for 4-R tau and 3-R tau. The AD component will have both 3-R and 4-R tau, whereas the astrocytic elements of ARTAG should only contain 4-R tau. In the final report, the presence of ARTAG should be mentioned in addition to any other pathology. 

#### 2.1.3. Argyrophilic Grain Disease (AGD)

Argyrophilic grain disease (AGD) is a sporadic tauopathy that is seen in the elderly and the differential diagnosis of late onset cognitive impairment without established dementia [19,20]. Clinically, the entity is ill-defined, although some series have reported anxiousness, restlessness and depression. The characteristic features are the so-called argyrophilic grains, seen in the amygdala, insular cortex, entorhinal/transentorhinal cortex, presubiculum and the CA1 segment of the hippocampus. As well as the Gallyas silver stain, p62, and especially 4-R tau are especially good at detecting grains, as it is thought to be a 4-R tauopathy. Other pathologies such as pretangles, neuropil threads and oligodendroglial inclusions are sometimes evident. Usually, there are no neuritic plaques and only very little Aβ deposition is present, but, rarely, grains can be associated with AD changes. The presence of HP-tau positive pretangles in the dentate fascia may aid the diagnosis. Whether AGD represents a distinct entity or may lower the threshold of dementia is a question that still remains to be adequately answered. 

##### Practical Considerations 

Again, it is extremely useful have an indication of the age of the subject and the degree of cognitive decline as well as other symptoms. The dot-like and comma-like grains can be seen on HP-tau immunohistochemistry (Figure 4a,c) and they have different configurations from threads and neurites. Nevertheless, in the background of other tauopathies including AD they may be difficult to pick out. Again 4-R and 3-R tau immunohistochemistry can prove useful, as the grains are 4-R tau immunopositive (Figure 4b,d) but 3-R tau immunonegative, unlike AD, where the threads and neurites are a mixture of 3-R and 4-R tau. In the final report, the presence of grains should be mentioned as it may be the determining factor of the cognitive decline or, in the presence of another pathology, it may have exacerbated the effects of those diseases. 

#### 2.1.4. Chronic Traumatic Encephalopathy (CTE) 

Chronic traumatic encephalopathy (CTE) is a clinicopathological entity which used to be termed dementia pugilistica before it was noted not to be confined to boxers. It has been seen in American and Association football players as well as wrestlers and it is thought to result from repeated concussive or subconcussive blows to the head rather than a solitary severe head injury [21,22,23]. Clinically, it is associated with aggression, irritability, short-term memory problems, depression and suicidal tendencies. There may be cerebral atrophy (Figure 5a,b) and CTE is considered primarily a tauopathy occurring 8–10 years after repetitive head trauma, but it sometimes has additional TDP-43 and or α-synuclein pathology. There are neurofibrillary tangles and astrocytic fibrillary deposits present. The tangles are often irregularly distributed in the cortex and especially superficially, whereas the astrocytic tau pathology is concentrated in subpial and periventricular areas. Tau pathology is also seen at the depths of sulci (Figure 5c) and around vessels (Figure 5d). There can be some confusion and pathological overlap with AD; however, the distribution of tau pathology above is different from AD and there may be little associated Aβ pathology. Furthermore, there is often sparing of the calcarine cortex, unlike the higher stages of AD. Nevertheless, the biochemical composition of the HP-tau is identical to AD; however, recent work has found differences in tangle structure on electron microscopy [24]. Another similarity exists between the pathology of CTE and ARTAG with similar distributions, but ARTAG is considered to be primarily a 4-R tauopathy, whereas CTE has both 3-R and 4-R tau glial inclusions [18]. A staging system of I-IV has been suggested by McKee and colleagues [21].

##### Practical Considerations 

The most important issue here is to have a thorough clinical history so that one can determine if the subject suffered repeated head trauma and to what degree. The results of scans can be extremely useful. On macroscopic examination of the brain, it is particularly important to look for other features of trauma such as contusions/lacerations or haemorrhage. There may be other less specific indicators such as a caved or thinned or torn septum pellucidum (Figure 5b). Histologically, the distribution of tangles and other tau pathology within the depths of sulci are the key diagnostic factor together with the degree of Aβ pathology. If there is difficulty distinguishing ARTAG type pathology from CTE pathology, 3-R and 4-R tau immunohistochemistry can be attempted since the astrocytic tau pathology seen in ARTAG should be 4-R tau, whereas in CTE in should be a mixture of 3-R tau and 4-R tau. Nevertheless, making the diagnosis of CTE in the midst of AD and ARTAG pathology in an older person could be extremely difficult. Indeed, the concept of CTE and its defining pathological features are not universally accepted. The final report should discuss the possibility of CTE if the pathology suggests this, but only if there is some independent history of repeated head trauma. 

#### 2.1.5. Other Tauopathies-Pick’s Disease (PiD), Progressive Supranuclear Palsy (PSP), Corticobasal Degeneration (CBD), Globular Glial Tauopathy (GGT), Microtubular-Associated Protein tau (MAPT) Mutations 

It is less likely that PiD, PSP, CBD, GGT or MAPT mutations would be confused with AD pathologically. They are often associated with distinctive clinical syndromes. Some, such as PiD and GGT, very often present with a frontotemporal dementia (FTD), whilst PSP is associated with parkinsonian and and/or ocular disturbances and/or frontal lobar syndromes, CBD is associated with apraxia, cortical sensory signs parkinsonism and alien limb phenomena. MAPT mutations can be more variable depending on the mutation, but often manifest themselves with parkinsonism accompanied by behavioural and personality disturbances. Furthermore, they all, apart from MAPT mutations, can usually be diagnosed by the detection of almost exclusively 4-R tau in PSP, CBD and GGT and 3-R tau in PiD, together with the distinctive pathology of astrocytic plaques in CBD, globose neurofibrillary tangles in the basal ganglia and brainstem in PSP, globular glial white matter inclusions in GGT and Pick bodies in PiD. The pathology of all these conditions are described in more detail by Kovacs [25].

## 3. Alzheimer’s Disease and Additional Pathology 

This section deals with the challenges of neuropathological diagnosis when AD pathology is seen in conjunction with other pathological entities. It also aims to illustrate the measures that have been taken in an attempt to put into context the different pathologies and their likely contribution to the overall clinical picture. 

### 3.1. Alzheimer’s Disease and Lewy Bodies 

Lewy body dementias (LBDs) are the second commonest cause of dementia in the West and they are divided into two main types: Dementia with Lewy bodies (DLB) and Parkinson’s Disease Dementia (PDD). They share the core clinical features of cognitive decline, parkinsonism (required by PDD but not always seen in DLB), fluctuating level of cognition and visual hallucinations. The two entities, are, however, separated by the somewhat arbitrary “one year rule”. If dementia occurs in the background of established Parkinson’s disease at least one year after the onset of parkinsonism, it is called PDD. If, however, dementia precedes or occurs at the same time as the onset of the parkinsonism, it is called DLB. Despite the separation, Parkinson’s disease (PD), PDD, and DLB share neuropathological hallmarks—the neuronal cytoplasmic Lewy body (LB) and the Lewy neurite, both composed of an abnormal α-synuclein protein. The progression of the LBDs based on the work of Braak et al., and McKeith et al. is assumed to follow a caudal-rostral model, in that the earliest stages are seen in the brainstem then the limbic regions and finally the neocortex (Table 3) [26,27,28]. There are marked challenges in distinguishing PDD from DLB neuropathologically, although there is evidence suggesting that the neocortical pathology tends to be higher in DLB and striatal pathology (i.e., associated with motor symptoms) predominates in PDD. [29,30]. Lewy bodies are a common feature in AD cases, and indeed AD pathology is often seen in cases otherwise characterised pathologically as DLB or PDD. It has also been found that DLB cases exhibit associated AD –related pathology (both Braak stage and Thal phase) at a higher stage than PDD cases [31,32]. There is also substantial evidence to suggest that ApoE ε4 allele status is a risk factor not just for AD but also for DLB (and, to a lesser degree, PDD) [33,34]. Therefore, there appears to be an association between the pathological processes in the two neurodegenerative disease groups. Indeed, there would appear to be evidence of a synergy between tau and α-synuclein, but only locally [35,36]. The presence of AD-related pathologies shortens the survival rate, although it is difficult to ascertain the relative contributions of the α-synuclein, HP-tau, and Aβ to the clinical course [37,38,39]. Irwin et al., reported that 18% of pathologically confirmed cases of DLB did not develop a motor disorder but the majority did have co-existing intermediate or high AD neuropathological change [40]. Therefore, it may be that LB formation in the limbic system and neocortex is triggered or exacerbated in vulnerable cerebral regions already burdened with Aβ and HP-tau pathology, rather than following the expected caudal-rostral progression. This is supported by the suggested staging system of LBDs proposed by Beach et al., which argues against a strict caudal-rostral proposed by Braak and McKeith [41]. The picture is complicated further, therefore, by the findings that patients with a co-morbid AD with a synucleinopathy have more severe motor impairment and lower scores on dysexecutive and visuospatial function tests than AD alone [36]. Other studies have agreed and also added sleep behaviour, increased anxiety and appetite problems and sometimes hallucinations [42,43], whilst some have not found this difference [44]. 

#### 3.1.1. Amygdala Lewy Bodies in Alzheimer’s Disease

Uchikado et al., discovered that, in a series of 347 cases of AD, 87 cases had brainstem, transitional, or diffuse Lewy body disease. Of the remaining 260 cases, 62 (24%) had Lewy bodies, mainly confined to the amygdala [45]. The burden of α-synuclein pathology in this group was significantly less than in cases of AD/DLB or AD/PDD. This suggested that AD with amygdala Lewy bodies may be a separate entity from these other conditions and may be considered a variant of AD. This has been supported by work from Sorrentino et al., who discovered that the α-synuclein pathology is different when confined to the amygdala to when it is seen elsewhere, specifically the abundance of the aggregation prone carboxy truncated forms of the protein [46]. 

#### 3.1.2. Practical Considerations

Considering that the association between AD and LB pathology is relatively common, it is extremely important to detect any co-pathology in neuropathological examination as it may explain unusual clinical presentations. Clues to the possible presence of LBs in a case of AD may come in the clinical history with some parkinsonian features or visual hallucinations. Figure 6a–e illustrates a case of AD-modified Braak stage IV (intermediate level of AD neuropathological change according to NIA/AA criteria) and DLB (diffuse neocortical McKeith Lewy body type, Braak Lewy body stage 6). Macroscopically, there may be loss of pigment in the substantia nigra (Figure 6a) in addition to that in the locus coeruleus, which is also commonly seen in AD. LBs (especially in the brainstem) can usually be seen on H&E stains (Figure 6b). Block selection for α-synuclein is important. Medulla and/or pons plus midbrain (including nigra) is essential (Figure 6c), together with “limbic areas” such as cingulate gyrus and amygdala and neocortical areas such as the middle frontal gyrus (Figure 6d) and parietal lobe (Figure 6d inset). This would allow the detection of brainstem, limbic and/or diffuse neocortical LB pathology. An attempt to acquire a Braak and/or McKeith Lewy body stage should be made in addition to the AD stages [26,27]. It may not always be possible since in some cases it may “skip” some of the lower stages, as discussed above with reference to Beach et al.’s staging system [41]. If the LB pathology is confined to the amygdala (Figure 6f), this should be noted. All these features should be included in the report and, in addition to staging the relevant pathologies, there should, if possible, be some attempt at correlating the clinical manifestations with the final pathology. 

### 3.2. Alzheimer’s Disease with TDP-43 Pathology 

The transactive response DNA-binding protein (TDP-43) is an RNA- and DNA-binding protein. It is also a pathological hallmark of most cases of amyotrophic lateral sclerosis (ALS) in addition to frontotemporal lobar degeneration with TDP-43 inclusions (FTLD-TDP). Although ordinarily a nuclear protein, when it becomes pathologically misfolded it can form neuronal cytoplasmic inclusions, intranuclear inclusions and neurites. It was first discovered to be the main protein in cases of ALS and FTLD in 2006 [47,48]. However, soon after this discovery there were papers describing some of the pathological features in AD, Lewy Body diseases and CTE [49,50]. Typically, the inclusions were often seen in the limbic system, including the amygdala and the hippocampus, and did not usually extend into the neocortex. Often, there was associated hippocampal sclerosis. Even when involving the cerebral cortex in AD, it did not appear to follow the same pattern of spread as that in cases of FTLD-TDP, with confirmed fronto-temporal type dementing symptoms. Josephs et al. noted the pathology in particular in a very elderly subgroup of AD cases and discovered an apparent worsening of cognitive decline when the TDP-43 was present, which was later confirmed by other studies [51]. In view of this, Josephs and colleagues devised a staging system where the earliest TDP-43 pathology seen in the context of AD was in the amygdala (Figure 7a), then the entorhinal cortex and subiculum, and then the dentate gyrus (Figure 7b) or occipitotemporal gyrus (stage III), and then the inferior temporal gyrus (stage IV), and finally the middle frontal gyrus (stage V), of a different pattern and density than typical FTLD-TDP [52]. This staging system was later refined into six stages (Table 4) [53]. Further evidence of a difference of AD with TDP-43 inclusions from FTLD-TDP was supplied in the work of Tomé et al., who have shown that the type of protein inclusion associated with AD is often different to that associated with FTLD, and furthermore the C terminal species of TDP-43 seen in the AD cases correlated with AD symptoms, whereas the full-length species of TDP-43, when present, correlated more with FTD (frontotemporal dementia) symptoms [54]. More recently, there has been a move to give this predominantly age-related TDP-43 pathology a designation of its own as a Limbic predominant age-related TDP-43 encephalopathy (LATE). This sought, in some respects, to separate it from the associated AD pathology seen in many cases [55]. LATE-NC was the designation given to LATE with neuropathological change. The recommended grading was divided into three: (1) amygdala only, (2) hippocampus involvement and (3) middle frontal gyrus (Table 4). It was often associated with hippocampal sclerosis. However, the question remained as to the boundary zones between LATE-NC and FTLD-TDP. When compared with FTLD-TDP, LATE-NC has a later age of onset, and limbic predominance of neuropathological change [56]. Clinically, LATE-NC was more associated with an amnesic cognitive syndrome rather than the behavioural or aphasic syndromes more typically associated with FTLD-TDP [56]. It had also been established that the cognitive impairment was independent of other co-existing pathologies [57,58]. Like AD itself and LBD, this entity with localised TDP-43 pathology also correlates with the presence of the APOEε4 allele [59,60]. Occasionally, the TDP-43 pathology has been seen in cognitively unimpaired subjects, and this has been interpreted as its being in a preclinical phase, although this, of course, is a somewhat circular argument. When comparing “pure” LATE-NC (i.e., without severe comorbid pathologies) with “pure” Alzheimer’s disease regarding neuropathological change, those with the “pure” LATE-NC have a more gradual clinical decline [61]. Those cases with combined AD neuropathological change and LATE-NC show faster decline than either pure AD neuropathological change or “pure” LATE-NC. [51,57,58]. 

It is not uncommon to have AD pathology together with both additional Lewy Body pathology and TDP-43 pathology. Here, it is often even more difficult to allocate particular clinical signs and symptoms to specific pathological features. Bayram et al., showed that both the LBs and TDP-43 pathology were associated with advanced AD. TDP-43 was considered to be associated with increased aberrant motor behaviour, while LBs were associated with increased anxiety, sleep disturbances and appetite problems [42], which is somewhat different to those findings attributed to the separate pathologies, as noted above. It was also noted that the neuropsychiatric effects in both comorbidities were more pronounced when associated with the earlier stages of AD [42]. 

#### 3.2.1. Practical Considerations 

The subject may be very elderly and there may be severe hippocampal atrophy on macroscopic examination of the brain, but there are no specific macroscopic features that give clues of a diagnosis of AD with TDP-43-positive inclusions as distinct from pure AD. It is obviously important when considering the co-pathology that one stains some of the blocks taken for the AD diagnosis (such as amygdala, hippocampus (including dentate gyrus) and middle frontal gyrus) for phosphorylated (p)-TDP-43 in addition to Aβ and HP-tau. Obviously, the co-pathology of TDP-43 with LBs and AD would be detected by staining for α-synuclein, as indicated above. The final report should indicate the presence of TDP-43 pathology and its location. An attempt at Josephs and/or LATE-NT staging should also be made. If possible, there should be some attempt to correlate the clinical features with the pathology. Distinct behavioural problems in life may indicate FTD clinically, and therefore it may important to sample the brain more extensively (e.g., additional temporal and frontal lobe regions and parietal lobe) to rule out co-existing FTLD-TDP. 

### 3.3. Alzheimer’s Disease and Vascular Dementia 

It is not uncommon in neuropathology to see cerebrovascular disease (CVD) in association with AD pathology. Depending on how it is measured, it is one of the most common concurrent pathologies seen in AD. In some studies, 25–80% of demented subjects show both AD and cerebrovascular lesions [62]. It has been known for many years that CVD without concurrent AD pathology can lead to a form of dementia, and this is called Vascular dementia (VaD). The most common types of VaD are: 

A. Multi-infarct dementia which, as the name suggests, occurs when there are a number of cerebral infarcts over time, and which sequentially lead to reduced cognitive ability. There is often associated atherosclerosis of major intracranial or extracranial arteries or evidence of thrombo-emboli [63]. The areas of infarct themselves are usually readily identified at macroscopic examination of the brain (Figure 8a). In addition, they may be multiple, with both macroscopic and microscopic features, indicating different ages for the infarcts (Figure 8b). Early studies revealed that loss of more than 100mls of brain tissue caused dementia [64], although later studies have shown much less tissue loss is required to produce the symptoms [65]. However, some groups have found no direct correlation between tissue loss and worsening dementia scores [66]. Instead there have been suggestions that the number of lesions or the location of the lesions may be more important (see strategic infarct dementia below);

B. Subcortical vascular encephalopathy (previously called Binswanger’s disease). This is a condition involving demyelination, axonal loss in the deep white matter and lacunar infarcts in the white matter, basal ganglia and thalamus. This is often due to thickening in the walls of small blood vessels (or arteriolosclerosis) with or without occlusion. The result is chronic hypoperfusion and disturbances in cerebral blood flow. Macroscopically, the brains of affected patients show small lacune-like structures in the basal ganglia (and/or thalamus) and evidence of small infarcts (Figure 8c) or granularity (also known as leukoaraiosis) of the cerebral white matter. Microscopically, in addition to lacunar infarcts in the deep grey nuclei, there is often evidence of the rarefaction of deep white matter, an increase in perivascular spaces (Figure 8d) and thickened vessel walls (Figure 8d inset). There is evidence from Esiri et al. that the white matter changes and microinfarcts (together with cerebral amyloid angiopathy (see below)) are particularly important in invoking dementing symptoms—indeed, apparently more so than larger infarcts [67]. Some rare cases are associated with cerebral autosomal dominant arteriopathy with subcortical infarcts and leukoencephalopathy (CADASIL) and cerebral autosomal recessive arteriopathy with subcortical infarcts and leukoencephalopathy (CARASIL);

C. Strategic infarct dementia. This is a condition where solitary or multiple small infarcts in particular brain areas such as thalamus, hippocampus or caudate nucleus result in a dementing syndrome. Macroscopically and microscopically, there will be evidence of infarcts in the specific region(s). 

#### 3.3.1. Cerebral Amyloid Angiopathy (CAA)

Another specific cause of infarcts, especially microinfarcts, lacunar infarcts, white matter changes and sometimes large lobar haemorrhages, is cerebral amyloid angiopathy (CAA). Apart from very rare familial types, for example the British, Danish and Icelandic type of CAA, the vast majority of cases are associated with Aβ deposition in blood vessel walls in the leptomeninges and cortex. The condition is often associated with AD; in the Mayo Clinic Brain Bank 13% of cases of proven AD also had moderate to severe CAA [68], and it is estimated that 85–95% of cases of AD have at least evidence of some CAA. The Aβ deposited in the vessel walls is not identical to that seen in plaques; whereas the parenchymal plaques are predominantly composed of the Aβ42 species, the deposits in CAA are mainly Aβ40. The CAA is sometimes suspected macroscopically when there is some orange discoloration of the leptomeninges, and the parietal and occipital lobes are more vulnerable than the temporal and frontal [69]. CAA is readily detected on immunohistochemistry for Aβ or amyloid stains, and they are divided into two main types: type 1 is considered the more severe, where there are capillaries showing CAA in addition to larger calibre arteries and arterioles, and type 2, where arteries and arterioles are involved but there are no affected capillaries [70] (Figure 8e,f)). They are also graded as mild, moderate and severe according to Vonsattel’s grading system [71]. Occasionally, there is an amyloid beta related angiitis (ABRA) associated with the CAA.

Because of the common association of AD with VaD, there have been attempts to clarify the relationship between the two and the relative importance of the individual pathological features to the clinical presentation and degree of the dementing syndrome. Love and Miners, in their review, discussed the possible inter-relationship of AD and VaD [72]. Both severe CAA and AD are associated with APOEε4 positive patients [73]. There is also evidence to suggest that cerebral hypoperfusion is an early feature of AD and probably accelerates the progression of AD. Gold and colleagues attempted to discover the neuropathological thresholds of Alzheimer’s disease pathology and vascular lesions [74]. By using a semi-quantitative scoring method, they found that the scores of Braak neurofibrillary tau (NFT), stages of cerebral microinfarct (CMI) and thalamic and basal ganglia lacunes (TBGL) were related to the clinical dementia rating scale, whereas other features such as white matter lacunes, white matter demyelination scores as well as gliosis scores were not related [74]. They therefore proposed that those demented cases with a Braak NFT stage >II and CMI+TBGL score > 2 may be classified as having mixed dementia, whereas those cases with Braak NFT stage ≤II and CMI+TBGL score of >2 should be considered as pure VaD and those cases with Braak NFT>II and CMI+TBGL ≤ 2 charactersised as pure AD [74]. However, their cohort did exclude cases of larger cerebral infarcts, and therefore was not fully comprehensive. A further study into the relationship of cognitive impairment and cerebrovascular pathology was performed by Skrobot et al. [75]. By employing a multi-stage study including the reproducibility of scoring, and validation by multiple neuropathologists, they assessed multiple neuroanatomical sites in 113 cases. They discovered three pathological features which correlated with cognitive decline. These were large subcortical infarcts of greater than 10mm in diameter, moderate or severe occipital leptomeningeal CAA, and moderate or severe occipital white matter arteiolosclerosis. The infarct was calculated as the most important of the three and was given greater weight. This was called the vascular cognitive impairment neuropathology guidelines (VCING) model. It then gave a score of low, moderate or high likelihood that cerebrovascular disease contributed to cognitive impairment in an individual case (see Table 5) [75]. Although it did not mention co-pathologies such as AD in the methodology, it can be used in conjunction with the NIA-AA guidelines for AD diagnosis to give an indication of the likely contributions of each respective disease to the overall cognitive decline. This is a relatively recent model and needs further validation. 

#### 3.3.2. Practical Considerations 

When considering whether there is any significant CVD to give a diagnosis of VaD, one needs to carefully examine the clinical history. Stepwise deterioration in cognition may indicate VaD. At macroscopic examination, one should seek out foci of orange discoloration in the leptomeninges, (which may indicate CAA or small haemorrhages), any areas of infarction (or haemorrhage) and their approximate age. It is noteworthy that in many Brain Banks, including the London Neurodegenerative Diseases Brain Bank at King’s College London, the brain is often halved and one hemisphere is fixed in formalin for neuropathological examination, whist the other is sliced fresh, selected blocks taken, and then these blocks and this hemisphere is frozen. In such cases, careful examination of this fresh hemisphere needs also to be undertaken since CVD can be in either hemisphere and can be asymmetrical. Other observations on the fixed hemisphere or brain should include the presence of lacune-like infarcts in the basal ganglia/thalamus and any rarefaction or granularity to the white matter. These features can indicate subcortical vascular encephalopathy. The blocks taken should include any areas of infarction, and areas of deep white matter (especially the occipital white matter) and basal ganglia (and/or thalamic blocks). In addition to the common H&E stain, myelin stains such as Luxol Fast Blue/Nissl (LFB/N) can be employed to better visualise the infarct or loss of myelin. Aβ immunohistochemistry, hopefully already used on the frontal/temporal and occipital lobes, can give an indication to the type (1 or 2) and severity of any CAA present. The report should indicate the type and extent of any CVD present. An attempt to evaluate the likelihood of the lesions contributing to dementing symptoms can be attempted by applying the VCING protocol. Therefore, some attempt at clinico-pathological correlation can be made and, if possible, when seen in conjunction with AD pathology, the relative likely contributions of the pathologies to the clinical picture. 

### 3.4. The Combination of Alzheimer’s Disease and More Unusual Pathologies 

The combinations described above are not exhaustive and there are rarer occasions when AD pathology can be seen in combination with other pathological entities such as ALS, PSP, CBD, Multiple System Atrophy (MSA), Huntington’s disease (HD), prion disease, multiple sclerosis and cerebral tumours. These can only be evaluated by attention to the clinical notes as well as careful macroscopic examination, extensive brain block selection and a wide range of immunohistochemical stains. 

## 4. Post- Mortem Brain Block taking Protocol and Pathology Table 

### 4.1. Block Taking Protocol

An example of the block taking and staging schedule is given in Table 6. It is the schedule that is currently employed at the London Neurodegenerative Diseases Brain Bank at the IOPPN at King’s College London. This particular table is used for a suspected case of AD and it is modified slightly if there is good clinical indication of a different type of neurodegenerative disease, e.g., ALS or HD. It can be seen that, from the perspective of AD diagnosis, it allows the full analysis of a modified Braak (BNE) stage, a Thal Aβ phase and a CERAD assessment of plaque density, therefore giving an ABC score, and hence a level of AD neuropathological change. It also allows for detecting AD mimics such as PART, ARTAG, and AGD. Other stains such as 4-R and 3-R tau can be added if needed. The staining for α-synuclein allows detection of LBs and the possibility of a Braak or McKeith LB stage. Furthermore, the p-TDP-43 immunohistochemistry on selected blocks allows detection of the abnormal protein in limbic and neocortical areas. Finally, the white matter blocks and the Aβ immunohistochemistry on the occipital lobe allow a VCING stage to be given. Overall, therefore, using this schedule allows the vast majority of the pathological entities discussed in the previous sections to be detected. Other stains and blocks can be added if necessary for individual cases. For example, occasional cases of previously unsuspected prion disease are encountered and additional immunohistochemistry for prion protein can be performed whilst conforming to Health and Safety guidelines. The thalamic and motor cortex blocks are not routinely stained but are sampled, because sometimes other clinical details only become available after the initial neuropathological examination. 

### 4.2. Pathology Table for Research Requests 

In addition to providing patients’ families and clinicians with neuropathological diagnoses, the purpose of brain banks is to provide researchers with high-quality human brain material. It is obviously extremely important that this material is characterised neuropathologically as accurately as possible. Some years ago, the Brains for Dementia Research (BDR) brain bank neuropathologists devised a pathology table (Figure 9) which allows a semiquantitative assessment of the neuropathological loads of various pathologies in individual cases in anonymised form. It has now been taken up by the UK Medical Research Council (MRC) Brain Bank Network for their database. The white boxes seen in the table are the compulsory areas to be completed, whilst the blue boxes are optional, and all scores except the Aβ brain boxes (which is itself scored present: 1 or absent: 0) are scored (0: negative, 1: mild, 2: moderate, 3: severe). By way of dropdown boxes, the table records the relevant stages (or types) for AD, LBD, TDP-43 pathology and VaD, as well as recording features such as ARTAG and PART and giving a final diagnosis or diagnoses. It is only an approximate guide and has not been validated across several centres, and therefore is not available for general viewing on the MRC website, nevertheless, it is available to all the brain banks in the network and allows them to provide researchers with brain tissue containing more specific pathological features from particular anatomical regions. 

## 5. Discussion

The advent of reliable immunohistochemistry revolutionised the neuropathological diagnosis of Alzheimer’s disease. Rather than relying on silver stains, a more scientific approach could now be adopted when assessing the pathological patterns of Aβ and HP-tau. Combined with more extensive brain sampling and better clinical and radiological assessment came the realisation that there were other conditions, some age-related, that revealed HP-tau either in a different chemical composition to that seen in AD (such as AGD and ARTAG) or in a different neuroanatomical distribution from AD (such as CTE) or without the corresponding second Aβ pathology, as seen in AD (such as PART). Furthermore, the discovery of other important abnormal forms of proteins, such as α-synuclein and p-TDP-43, typically associated with Lewy body diseases and FTLD/ALS, respectively, added a different dimension again to the diagnosis of AD because of the recognition of relatively common concomitant pathologies. The purpose of neuropathological examination of post-mortem brains in cases of neurodegeneration including AD is twofold. Firstly, to provide a diagnosis for the patient’s next of kin and family so that they can have some appreciation of the reason for their relative’s distressing symptoms in life and/or on occasions to determine the likelihood of genetic inheritance; secondly, in specialised units, to provide researchers with brain (and spinal cord) tissue from neuropathologically well-characterised cases. The presence and recognition of age-related tau pathologies, whether PART, ARTAG or even AGD, either separately or in combination with AD pathology, may lead to more discerning clinical, biochemical and radiological evaluations of AD in the future. The recent clinicopathological correlative work on concomitant pathologies such as Lewy bodies, TDP-43 pathology or CVD with AD has begun to determine the likely interaction of the various pathologies, the underlying risk factors and the threshold for the respective pathologies in causing dementing symptoms. The fact that these relationships are beginning to be better understood is a welcome development that will hopefully lead to improved clinical diagnosis and to more awareness as to why specific anti-individual protein (such as anti-Aβ or Anti-HP-tau) clinical trials may have only limited success. It could, ideally, lead to more targeted individual therapies. 

## Figures and Tables

**Figure 1 brainsci-10-00479-f001:**
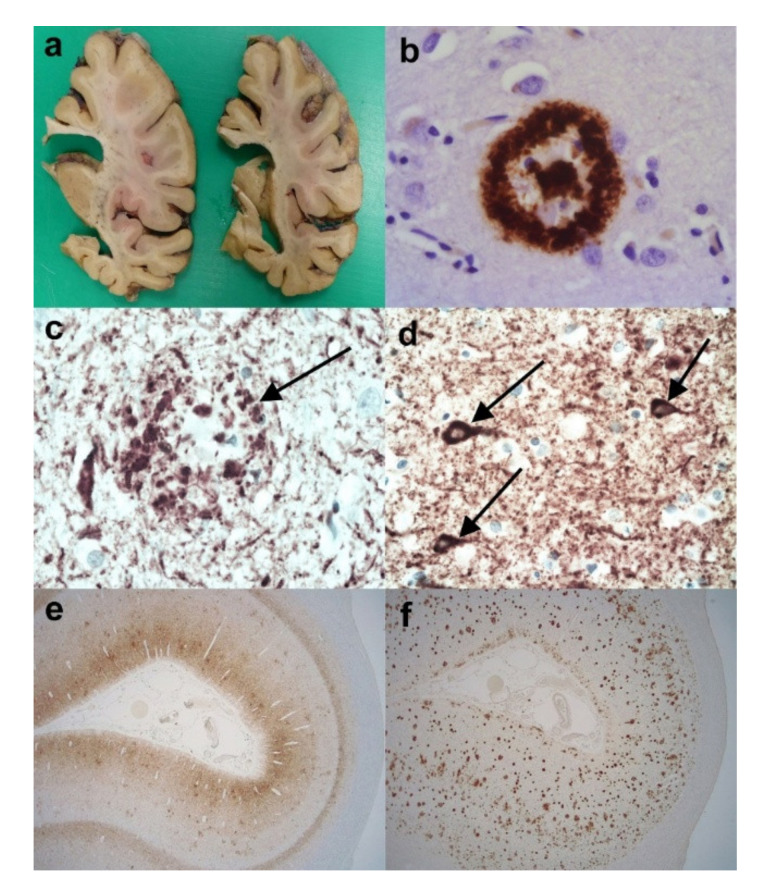
The neuropathology of Alzheimer’s disease (AD). (**a**) Coronal slices of right cerebral hemisphere from a case of Braak Stage VI AD. Note the degree of cerebral atrophy with dilated ventricles and small hippocampus. (**b**) An Aβ immunopositive cored plaque within the cerebral cortex. (Anti-Aβ). (**c**) An HP-tau immunopositive neuritic plaque (arrow) present within the cerebral cortex. Note the central non-staining region and the background high density of HP-tau immunopositive neuropil threads. (Anti-HP-tau). (**d**) HP-tau immunopositive neurofibrillary tangles (arrows) in a background high density of HP-immunopositive neuropil threads. (Anti-HP-tau). (**e**) Calcarine cortex showing strong immunopositivity for HP-tau made up of neurofibrillary tangles, neuropil threads and neuritic plaques. (Anti-HP-tau). (**f**) Calcarine cortex showing strong parenchymal immunopositivity for Aβ, mainly in the form of plaques. (Anti-Aβ). Original magnifications (**b**–**d**) x60, (**e**,**f**) x1.25.

**Figure 2 brainsci-10-00479-f002:**
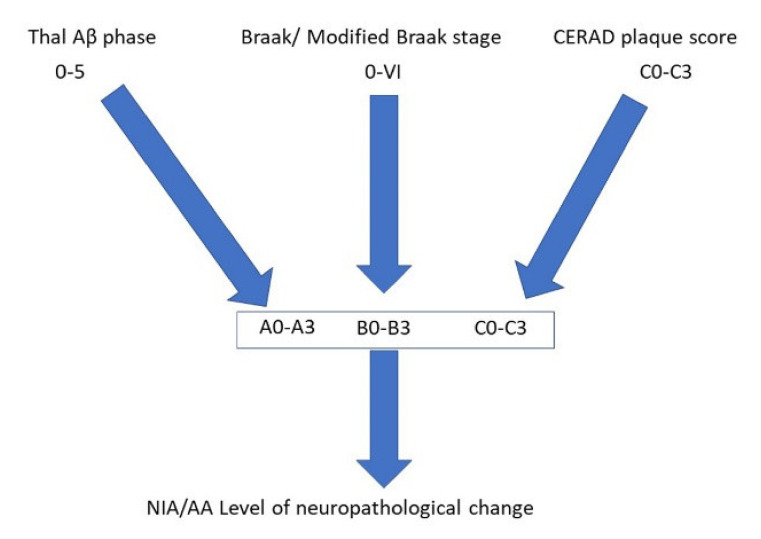
Pathway to illustrate the combination of different pathological features that allows an indication of the level of Alzheimer’s disease neuropathological change according to the National Institute on Aging/Alzheimer’s Association (NIA/AA) guidelines.

**Figure 3 brainsci-10-00479-f003:**
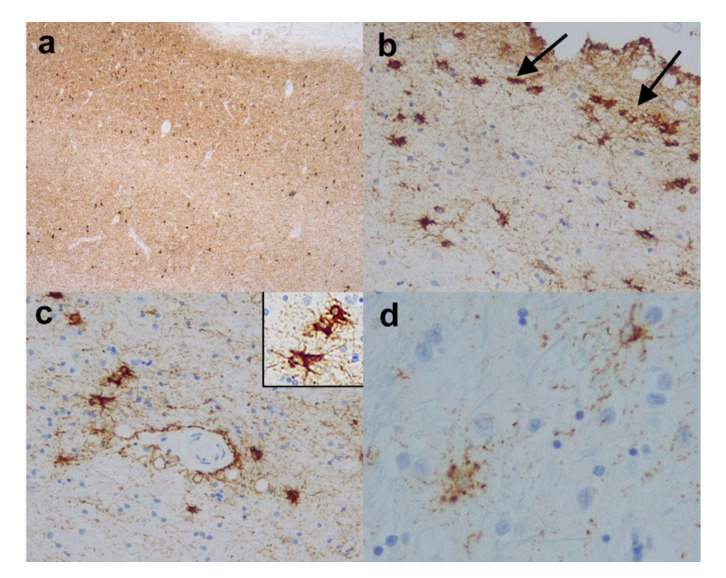
Age related tauopathies. (**a**) Entorhinal cortex from a case of primary age-rated tauopathy (PART) revealing strong immunopositivity for HP-tau in the form of neuropil threads and neurofibrillary tangles, but no neuritic plaques and there was very little corresponding Aβ pathology. (Anti-HP-tau). (**b**) Medial temporal (entorhinal) cortex and (**c**) white matter around amygdala from a case showing aging-related tau astrogliopathy (ARTAG). Note the superficial HP-tau immunopositive glial inclusions in (**b**) (arrows) and the perivascular HP-tau immunopositive thorny astrocytes in (**c**) and (**c**) inset. (**d**) Reveals granular/fuzzy astrocytes in the cerebral cortex from a case showing ARTAG. (Anti-HP-tau). Original magnifications (**a**) x4, (**b**,**c**) x10, (**c**) inset x20, (**d**) x40.

**Figure 4 brainsci-10-00479-f004:**
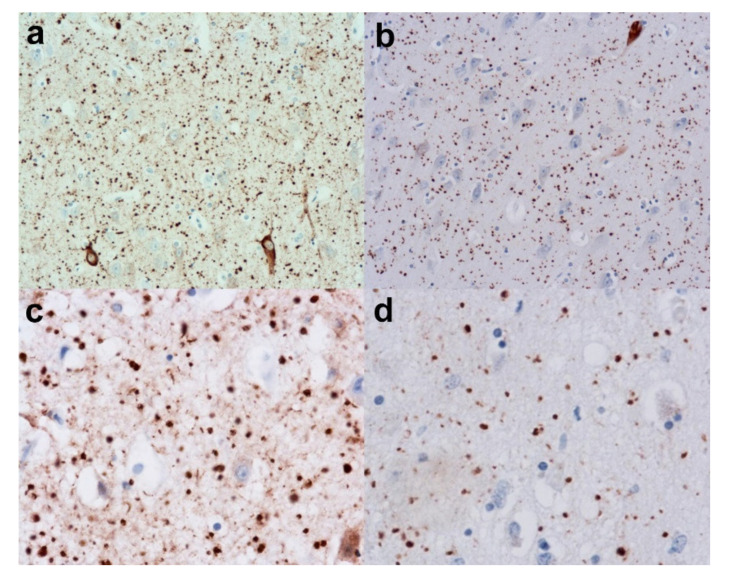
Argyrophilic Grain disease (AGD). (**a**) Reveals a low-power view of the grains in the form of dots in the hippocampus. (Anti-HP-tau). (**b**) Shows the grains to be immunopositive for 4-Repeat (4-R) tau. (Anti-4-R tau). (**c**) Reveals a higher power view of the grains (Anti-HP-tau) and (**d**) shows the strong immunopositivity for 4-R tau. (Anti-4-R tau). Original magnifications (**a**,**b**) x20, (**c**,**d**) x40.

**Figure 5 brainsci-10-00479-f005:**
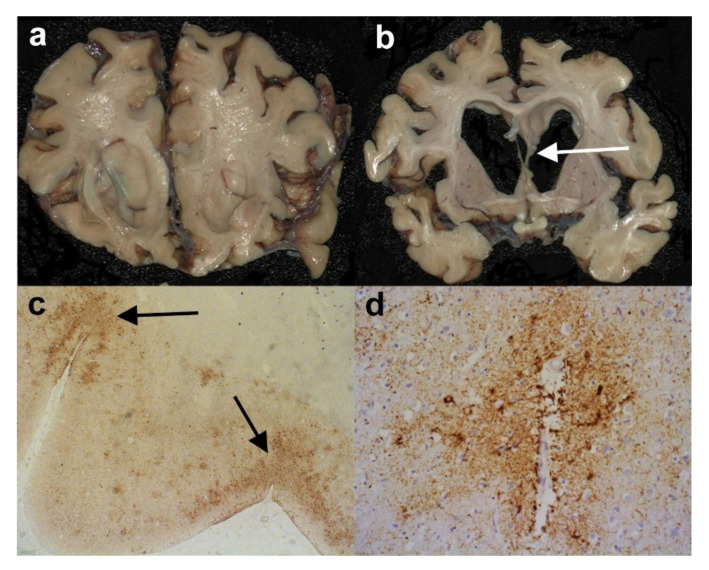
Chronic traumatic encephalopathy (CTE). (**a**) and (**b**) Coronal slices from the cerebral hemispheres of a man who had been an amateur boxer in life. Note the degree of cerebral atrophy and the thinned and partially torn septum pellucidum in (**b**) (arrow). (**c**) Sections from the parietal lobe where the highest density of HP-tau immunoreactivity is seen in the depths of sulci (arrows). (Anti-HP-tau). (**d**) There are also a number of HP-tau immunopositive tau glial inclusions in a perivascular arrangement. (Anti-HP-tau). Original magnifications (**c**) x2, (**d**) x10.

**Figure 6 brainsci-10-00479-f006:**
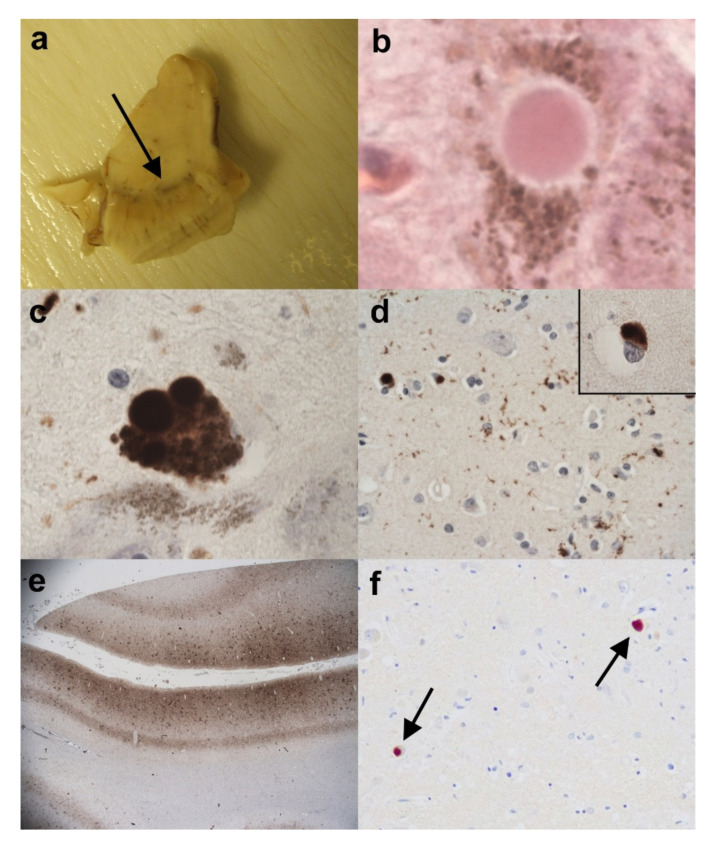
(**a**–**e**). A case of concomitant AD (Braak/BNE Stage IV) and DLB (diffuse neocortical stage). (**a**) Half midbrain showing pallor in the substantia nigra (arrow). (**b**) Lewy bodies are identified on H&E in nigral neurones. (H&E). (**c**) Lewy bodies are confirmed in the substantia nigra on immunohistochemistry for α-synuclein. (Anti-α-synuclein). (**d**) α-synuclein immunopositive Lewy bodies are also seen in the frontal neocortex (**d**) and in the parietal cortex (**d**) inset. (Anti-α-synuclein). (**e**) A low power view of the middle temporal gyrus revealing a high density for HP-tau. This is made up of neuropil threads, neuritic plaques and neurofibrillary tangles. (Anti-HP-tau). (**f**) A different case of AD with only occasional Lewy bodies only seen in the amygdala (arrows). (Anti-α-synuclein). Original magnifications (**b**–**d**) inset x60, (**d**) x40, (**e**) x1.25, (**f**) x20.

**Figure 7 brainsci-10-00479-f007:**
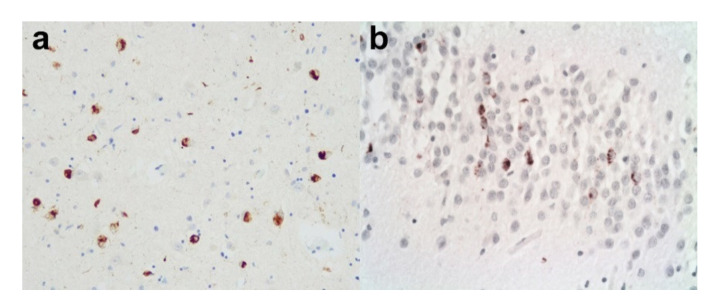
Limbic TDP-43 pathology. A case of AD with additional limbic TDP-43 pathology. (**a**) There are moderately frequent TDP-43 immunopositive neuronal cytoplasmic inclusions in the amygdala. (Anti-p-TDP-43). (**b**) There are also TDP-43 immunopositive neuronal cytoplasmic inclusions in the dentate gyrus. (Anti-p-TDP-43). Original magnifications (**a**,**b**) x20.

**Figure 8 brainsci-10-00479-f008:**
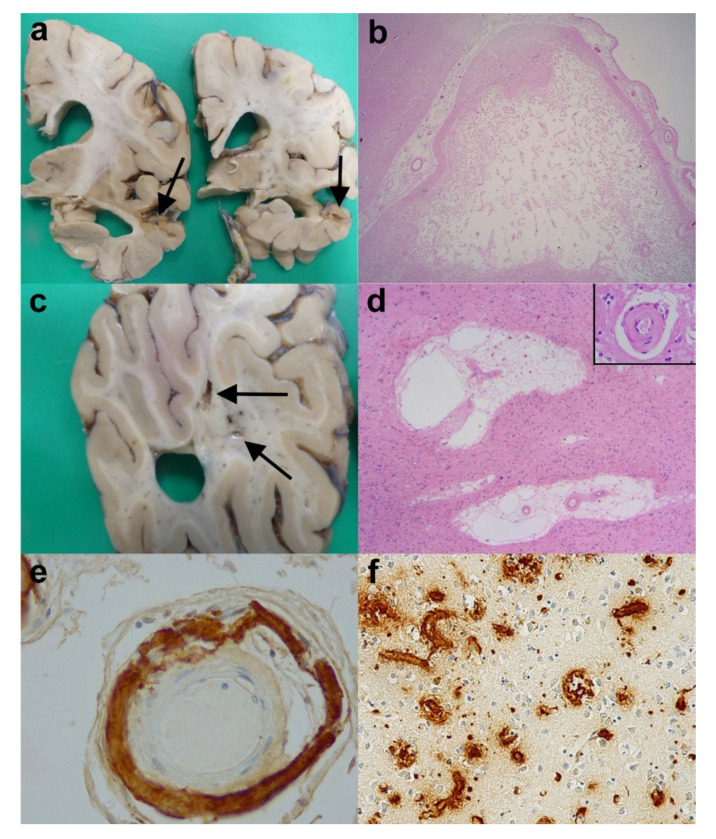
Cerebrovascular disease (CVD) and Vascular dementia (VaD). (**a**) Coronal slices of the right cerebral hemisphere from a case of AD and cerebral infarcts. Note the marked cerebral atrophy and the infarct (arrows). (**b**) An old cystic cerebral infarct in the temporal cortex (H&E).(**c**) A coronal slice from the right parieto-occipital lobe of a case of subcortical vascular encephalopathy showing small cystic infarcts in the cerebral white matter (arrows). (**d**) The subcortical vascular encephalopathy is characterised by rarefaction of white matter, increase in perivascular spaces and thickening to vessel walls (inset). (H&E). (**e**) Severe amyloid angiopathy within a leptomeningeal artery (Anti-Aβ). (**f**) Extensive amyloid angiopathy within parenchymal blood vessels including capillaries. (Anti-Aβ). Original magnifications (**b**) x1.25, (**d**) x2, (**d**) inset x40, (**e**) x20, (**f**) x10.

**Figure 9 brainsci-10-00479-f009:**
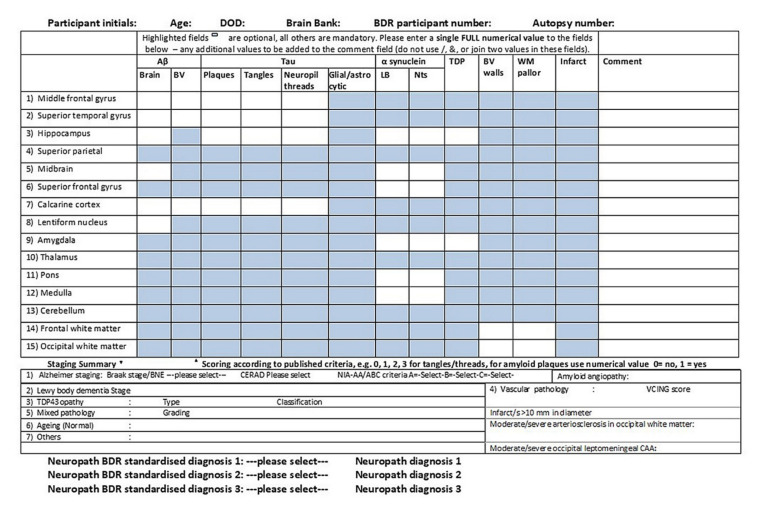
The neuropathology table used by the Brains for Dementia Research (BDR) Brain Bank Consortium to estimate regional pathological severity in individual cases.

**Table 1 brainsci-10-00479-t001:** Showing the translation of the Thal, Braak/BNE stages and CERAD plaque score into A,B,C for assessment of the level of AD neuropathological change (after Montine et al. [13]).

A- Aβ Plaque Score (Thal)	B- Braak/Modified Braak Stage (Braak and Braak/BNE)	C-Neuritic plaque Score (CERAD)
A0- No Aβ plaques	B0- No tangles or threads	C0-No neuritic plaques
A1-Thal phase 1 or 2	B1-Braak stage I or II	C1-CERAD score sparse
A2-Thal phase 3	B2-Braak stage III or IV	C2-CERAD score moderate
A3-Thal phase 4 or 5	B3-Braak stage V or VI	C3-CERAD score frequent

**Table 2 brainsci-10-00479-t002:** Showing the NIA/AA scheme for the level of AD neuropathological change for the various permutations of A,B,C scores (after Montine et al. [13]).

A-Amyloid (Thal)	C-Neuritic Plaque (CERAD)	Neurofibrillary Tangles (and Threads) Braak/BNE Stage)		
		B0 or B1 (0-II)	B2 (III/IV)	B3 (V/VI)
A0 (0)	C0	Not AD	Not AD	Not AD
A1 (1/2)	C0 or C1	Low	Low	Low
	C2 or C3	Low	Intermediate	Intermediate
A2 (3)	Any C	Low	Intermediate	Intermediate
A3 (4/5)	C0 or C1	Low	Intermediate	Intermediate
	C2 or C3	Low	Intermediate	High

**Table 3 brainsci-10-00479-t003:** A table comparing the Braak stage and McKeith type of Lewy body diseases (after Alafuzoff et al. [28]).

Brain Area	Medulla		Pons		Mid	Basal Forebrain		Hippocamp		Cing	Temp Cortex	Frontal Cortex	Parietal Cortex
Region	X	irz	LC	R	SN	nbM	Am	CA2	TOcx				
Braak stage	1	1	2	2	3	3	4	3	4	5	5	6	6
McKeith Type	←----------Brainstem----------→					←---------------------Limbic-------------------------------→						Neocortical	
Amyg type							Am						

Amyg-Amygdala, Mid-Midbrain, Hippocamp-Hippocampus, Cing-Cingulate, Temp-Temporal, X-Dorsal motor nucleus of vagus, irz- intermediate reticular zone, LC –locus coeruleus, SN-substantia nigra, Am-amygdala, R-raphe, nbM- nucleus basalis of Meynert, TOcx Temporo-occipital cortex.

**Table 4 brainsci-10-00479-t004:** Comparing the LATE-NC stages of TDP-43 pathology with those of Josephs (after Josephs et al., and Nelson et al. [53,55]).

LATE-NC		Josephs	
0	None	0	None
1	Amygdala	1	Amygdala
2	Hippocampus	2	Entorhinal cortex, subiculum
		3	Dentate, occipitotemporal cortex
		4	Insula, inferior temporal cortex
		5	Inferior olive, midbrain
3	Middle frontal gyrus	6	Basal ganglia, middle frontal gyrus

**Table 5 brainsci-10-00479-t005:** Illustrating the VCING Model estimating the likelihood that cerebrovascular disease contributed to cognitive impairment (after Skrobot et al. [75]).

Likelihood that Cerebral Vascular Disease Contributed to Cognitive Impairment	<---Low (<50%) --->			Moderate (50–80%)		<--High (>80%) --->		
One or more large subcortical cerebral infarcts	-	-	-	+	-	+	+	+
Moderate or severe occipital leptomeningeal CAA	-	+	-	-	+	+	-	+
Moderate or severe occipital white matter arteriolosclerosis	-	-	+	-	+	-	+	+

-not present, + present.

**Table 6 brainsci-10-00479-t006:** Illustrating the blocks routinely taken from fixed post-mortem brains and the stains employed in a suspected case of Alzheimer’s Disease.

Block Location	Stains
1. Middle frontal gyrus	H&E, Aβ, HP-tau, p62, pTDP-43
2. Superior and middle temporal gyri	H&E, Aβ, HP-tau, p62, pTDP-43
3. Hippocampus	H&E, Aβ, HP-tau, p62, α-syn, pTDP-43
4. Parietal lobe	H&E, HP-tau, α-syn
5. Mid-brain	H&E, Aβ, α-syn
6. Superior frontal gyrus and cingulate gyrus	H&E, α-syn
7. Occipital including calcarine and paracalcarine	H&E, Aβ, HP-tau
8. Basal Ganglia	H&E, Aβ
9 Amygdala	H&E, Aβ, HP-tau, p62, α-syn, pTDP-43
10. Thalamus	(No stains)
11. Pons	H&E, α-syn
12. Medulla	H&E, α-syn
13. Cerebellar hemisphere	H&E, Aβ, p62
14. Frontal deep white matter	H&E (LFB/N-if evidence of CVD)
15. Occipital deep white matter	H&E (LFB/N-if evidence of CVD)
16. Motor cortex	(No stains)

(No stains)-indicate block is taken and not routinely stained but may be if need arises. CVD-cerebrovascular disease, α-syn-α-synuclein, LFB/N-Luxol Fast Blue/Nissl.

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
