# Peer review of "The Neuropathological Diagnosis of Alzheimer’s Disease—The Challenges of Pathological Mimics and Concomitant Pathology"

_brainsci, 2020, doi:10.3390/brainsci10080479_

Round 1
Reviewer 1 Report
In the Review titled “The Neuropathological Diagnosis of Alzheimer’s 2 Disease – The Challenges of Pathological Mimics and 3 Concomitant Pathology” the authors describe current diagnostic methods to identify Alzheimer’s Disease (AD) patients and other concomitant pathologies with the aim of evaluating the effect of each pathology and how they affect the final clinical diagnosis.
The present review is very interesting because, currently, it is very difficult to discriminate the different pathologies with dementia because there are many pathologies that mimics the same symptoms. It is relevant for clinicians and also for researchers, since conducting a clinical trial requires an accurate patient diagnosis to be successful.
In my opinion, only minor revision should be done:
- Typo in legend of Figure 7: “d)” should be changed by b)
- Typo in title of Table 4: “TDD-43” should be corrected to TDP-43
- Legend of Figure 8: panel c) is missing in the legend.
- Table 5 needs to be better explained because it is not clear the meaning of each “-“ and “+”.
Reviewer 2 Report
The paper is well written and organized. It is an up-to-date, comprehensive review of the literature in the field of AD neuropathology. The "Practical Considerations" comments for each individual chapters are intriguing and innovative.
Critical points:
1) in many part it has a neuropathology textbook profile.
2) it lacks of criticism. At the end, it should be noted that immunohistochemistry although relevant for understanding the molecular mechanisms of neurodegeneration process has not improved AD diagnosis.
3) it is unclear to me whether the data reported in Fig.s 1, 3, 4, 5, 6, 7, and 8 are original or other studies.
Reviewer 3 Report
The article highlights the increasing difficulty in the histotopathological diagnosis of neurodegenerative diseases as well as the importance of immunohistochemical markers against specific proteins involved in these diseases. This is a thorough, well-documented review.
Reviewer 4 Report
This review is extremely well written...the topics are interesting, pragmatic and potentially impactful. The organization is clear and logical.
Round 2
Reviewer 2 Report
All points have been taken into consideration.
In the current form the paper has improved slightly